# Lab-Tec@Home: A Cost-Effective Kit for Online Control Engineering Education

David Sotelo *, Carlos Sotelo *, Ricardo A. Ramirez-Mendoza *, Enrique A. López-Guajardo, David Navarro-Duran, Elvira Niño-Juárez and Adriana Vargas-Martinez

School of Engineering and Sciences, Tecnologico de Monterrey, Eugenio Garza Sada 2501, Monterrey 64849, Mexico; enrique.alopezg@tec.mx (E.A.L.-G.); david.navarro@tec.mx (D.N.-D.); enino@tec.mx (E.N.-J.); adriana.vargas.mtz@tec.mx (A.V.-M.)
* Correspondence: david.sotelo@tec.mx (D.S.); carlos.sotelo@tec.mx (C.S.); ricardo.ramirez@tec.mx (R.A.R.-M.)

**Abstract:** It is widely recognized that a hands-on laboratory experience is useful in control engineering education. Herein, the students overcome the main gaps between theoretical knowledge and experimental setups. Nowadays, in times of crisis due to the COVID-19 pandemic, virtual and remote laboratories are emerging as primary educational resources. However, in virtual labs, the students are not exposed to real life issues (i.e., equipment problems, noise, etc.) while in remote labs, communication and connectivity problems arise (i.e., network security, synchronization management, internet speed, etc.). Henceforth, this work presents an unpublished educational project named Lab-Tec@Home, and the aim of this research is to expand the access of hands-on control education at the undergraduate level. Here, students easily assemble a cost-effective laboratory kit at home and use it on their own computing devices connected with the external MATLAB/Simulink™ application. Thus, students can test and validate theoretical concepts of control engineering such as: system model identification, and PID control design and test. The assessment results show that the proposed educational project enhances the learning experience and has outstanding positive feedback of more than 290 students who undertook massive flexible digital courses at Tecnologico de Monterrey. This makes the proposed educational project mainly suitable for control engineering courses.

**Keywords:** education; control engineering; COVID-19; flexible learning; innovative pedagogy; PID

## 1. Introduction

Control engineering concepts are considered a difficult topic to be introduced at the undergraduate level [1,2]. Here, laboratories play an important role in bringing the 'real world' into theoretical education [3]. However, considering that the whole world is currently going through a critical pandemic situation in COVID-19, virtual and remote labs have gained substantial popularity in education for distance learning courses [4]. Several research works in control education have been reported recently in terms of virtual and remote labs [5–10]. In [8], level, position and temperature e-learning processes are described in detail to be used in an Automatic Control Master course at the "Universidad Nacional de Educación a Distancia" (UNED), Spain. Moreover, collaborative support to virtual and remote laboratories for a control engineering course is described in [10]. Additionally, in [6] an integrated system is proposed to configure and access remotely via certain type of sensors, actuators and controllers at the University of Huelva, Spain. In [9], a ball and beam remote lab is implemented in an automatic control engineering class to train students in different advanced techniques such as robust, fuzzy, and reset control, and in [7], a virtual lab environment for a four coupled tank system is developed at the Pontifical Catholic University of Peru (PUCP). However, while the modern internet has made the provision of remote laboratories relatively straightforward, there are still many practical obstacles; for example, students must queue to gain access which leads to frustration and

disengagement [11,12], especially for large class sizes commonly carried out during the COVID-19 pandemic. Moreover, as it is well mentioned in [13], remote lab activities tend to be based on virtual/mathematical environments, and are pseudo-authentic experiences. For this reason, the students need to work with real laboratory equipment to develop valuable technical skills and learning [3,14,15]. A lab take-home kit allows students to combine the benefits of remote labs, but using real hardware connected on their own computing devices [3,13]. The following research works related to lab take-home kits in control education have been encountered in literature [3,13–16]. In [3], a thermal control kit is implemented for an Advanced Control Systems course at the RMIT University in Melbourne, Australia. Furthermore, in [13], the Department of Automatic Control and System Engineering at the University of Sheffield in the United Kingdom developed a take-home helicopter equipment using LabVIEW$^{TM}$ with MyDAQ$^{TM}$ as the controller. In [14], a portable laboratory device, Flexy2, is developed at the Slovak University of Technology in Bratislava, Slovakia. Furthermore, in [15], a motion control system with a current direct motor for the Laboratory of Mechatronics is developed in the University of Brescia, Italy, and in [16], a control system laboratory kit with Raspberry Pi$^{TM}$ is used in an Advanced Control course at the University of Illinois in Urbana-Champaign, United States. The take-home laboratory kit for teaching control engineering education is not a new approach [17–20]. However, it is a challenge for large class sizes to be carried out remotely at a lower cost.

For that reason, this paper presents an unpublished educational project named Lab-Tec@Home, in which students easily assemble a low-cost, versatile and safe laboratory kit that can be easily shipped anywhere in the world and can be used by students who do not have a formal laboratory space to use [17,21,22]. The cost of the complete kit is around $20 and the logistic challenge in Mexico is supported by the supplier company, Tecnologico de Monterrey, Mecatronium$^{TM}$. The open-source software Arduino$^{TM}$ is used to handle the problem of the real-time data recording [23–27]. Four factors distinguish Arduino$^{TM}$: it is small in size, it is portable and ready to be used, the software is freely available, and supporting educational videos and material are freely made available. Moreover, using the MATLAB/Simulink$^{TM}$ application, students can plot the real data to carry out the model parameter estimation. Hence, students can exploit the learned theoretical concepts such as: model system identification, PID control design and performance index parameters acquired remotely through different massive flexible digital (MFD) model control courses.

The MFD model's main goal is the development of concepts, competencies and transversal skills needed to solve a control engineering problem. This model, implemented at Tecnologico de Monterrey, follows a masterclass approach, in which the best professors nationwide, aided by a team of experts and teaching assistants, provide different teaching–learning experiences to the students to foster the development of competencies, provide constant feedback and improve the students' overall learning experience. Under the MFD model, the developed kit could be viewed as an active learning and experiential learning activity that engages and motivates the students while providing a low-cost experience that reinforces the concepts learned. Up until now, this project is carried out to more than 290 students in Mexico who have expressed overwhelmingly positive feedback in the final institutional survey (ECOA).

## 2. Materials and Methods

Online courses for engineering education were already well implemented for small groups using Skype and Zoom [4]. However, education from home amid the Coronavirus lockdowns turns into a difficult task for massive courses. Furthermore, the regular e-learning courses alone cannot provide adequate skills or knowledge regarding laboratory experiments or analysis of real data.

In March 2020, the Tecnologico de Monterrey started its learning model, MFD, whereby students can take massive classes via remote interaction. The MFD model is a hybrid model (face-to-face and digital) mainly based on challenge-based learning and flipped class-

rooms as core student-centered learning techniques that offer flexibility to the students and professors, while engaging the students with different active learning and experiential activities [28–30]. This fosters the understanding concepts and the development of competencies and transversal skills required to solve different engineering problems. The massive nature of the model (expected to reach more than 300 engineering students nationwide) combined by the restriction in face-to-face interaction during the COVID-19 pandemic led to different challenges to provide continuous academic activities, experiences, and timely feedback. However, different interest-driven and active learning activities could be designed to overcome these challenges. Students will learn at their pace, taking advantage of different technological and digital tools while internalizing the concepts and results obtained (metacognition) while deepening their learning with class discussions and feedback from the professors [31,32].

Within the teaching–learning methodologies used, the proposed lab kit is designed to face the restrictions of using the laboratories in campus and meet the objectives, allowing the students to develop different competencies and skills in control engineering. This is already implemented in massive MFD courses such as: process automation, control engineering and control system. Here, based on an attractive real context related to the effects of supplementary light-emitting diodes (LED) on greenhouses [33–39], and considering PID is one of the most important topics in control theory [40,41], the students must design and implement a control system to regulate the lighting conditions. See Figure 1 (and Appendix A). The goal of the proposed kit is not to faithfully reconstruct a specific real greenhouse, but to set up a replica of a greenhouse for tomato plants in which students can identify the components and variables involved in a feedback control loop.

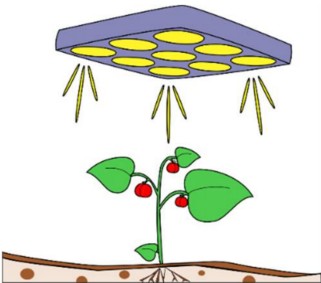

**Figure 1.** The effects of light-emitting diodes (LED) on a greenhouse.

The massive MFD courses are offered at different campuses of the Tecnologico de Monterrey such as: Monterrey, Guadalajara, Querétaro, Puebla, Toluca, Estado de México, Ciudad de México and Chihuahua. Even though more than 290 students are enrolled, they can work with their own kit at home. Figure 2 shows the influence of the lab kit in Mexico in a totally remote course.

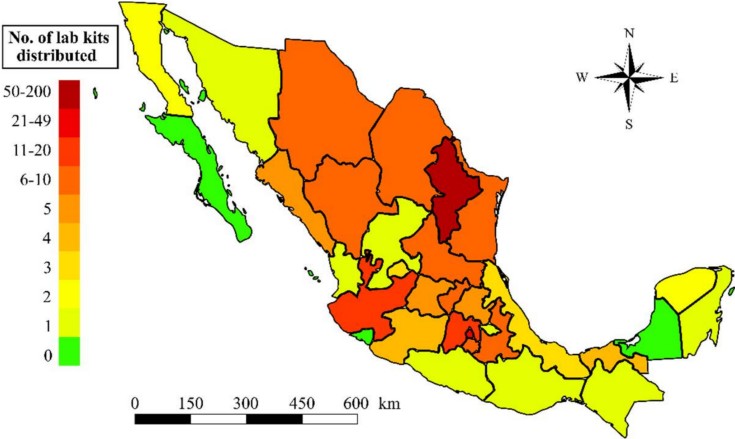

**Figure 2.** Distribution of lab kits in Mexico.

*2.1. Overall Course Methodology*

In Figure 3, the methodology followed by the students for the Lab-Tec@Home assessment is showed in solid lines, while the methodology followed by the professor for the overall approach, interventions and assessments, is represented in dashed lines. Both methodologies are integrated under the challenge-based learning (CBL) pedagogic technique (brackets). The overall objective of the course is to implement concepts and develop competencies required for control engineering topics. The project was implemented to assess the easiness of developing the kit without previous experience in electronics. Their metacognition process is conducted under a flipped classroom environment. This is categorized under the 'Big Idea' part. The course was offered nationwide from the summer of 2020 to August–December 2021. The following step is categorized as 'Research and Roadmap', in which the students undergo a pre-laboratory assignment and define the objectives and methodology. The students obtain information from validated sources needed to develop the model. During the 'Solution Proposal', learners start with the design cycle where the initial programming and prototypes are proposed to reach the main objective of the challenge. Here, the professor provides feedback, identifying different opportunity areas in the learning process. Thus, rebuttal questions about the control strategy are carried out and feedback is provided. Finally, the professor formulates an argumentative-driven assessment to evaluate the development of competencies.

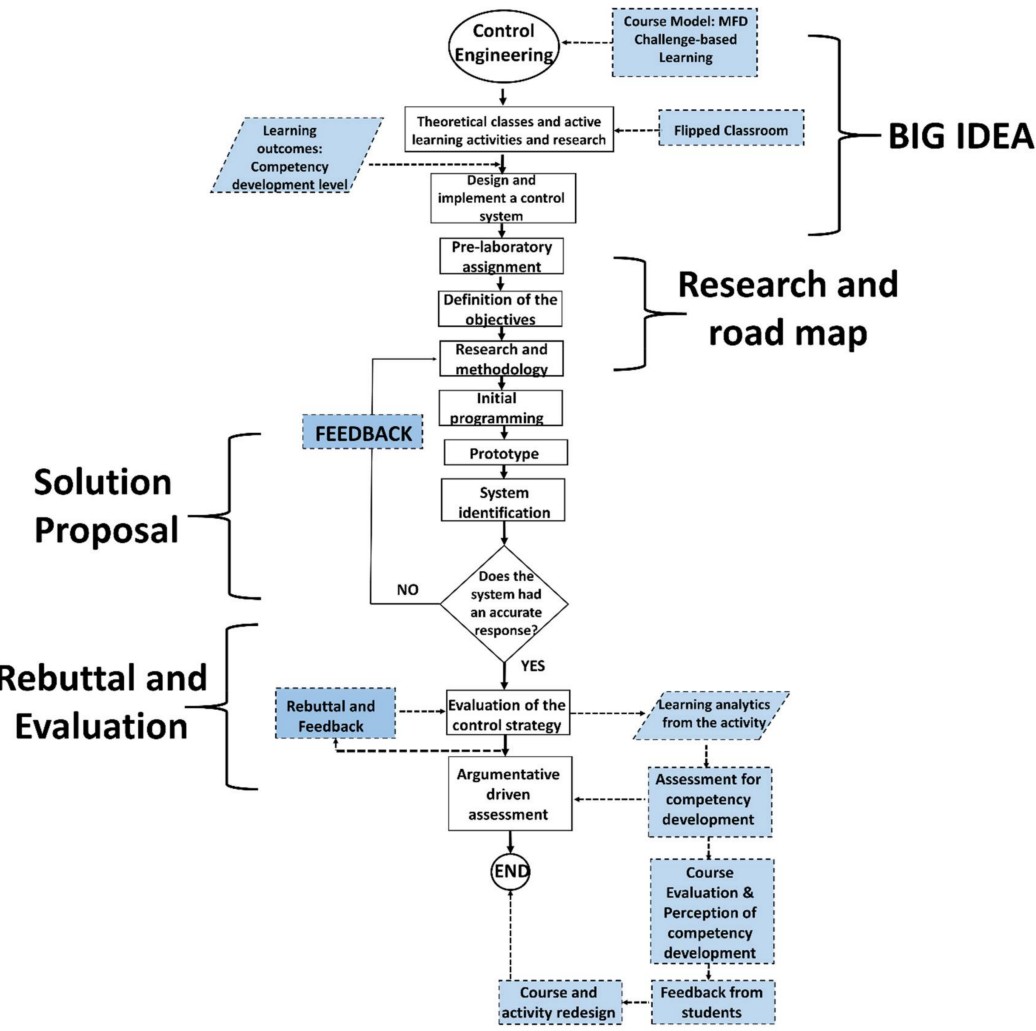

**Figure 3.** Course methodology for Lab-Tec@Home. The solid lines are the methodology followed by the students, while the dashed lines are the methodology and interventions of the professor.

### 2.2. Equipment Outline

The following five primary points are considered to develop the proposed kit:

1. Achieve control engineering educational objectives. Students must be able to design, implement and test the performance of different PID controllers.
2. Hardware must be largely plug-and-play. The control kit is compact and portable, and ready to be used by lecturers and students.
3. Low cost and accessibility of parts. The complete set is composed of electronic components that can be purchased by students and professors.
4. Safety for students and their computed devices. The lab kit is connected through the USB port; thus, the operating voltage is 5 [V] and the DC current per I/O Pin is 40 [mA], which does not represent a risk for users and equipment.
5. Portability and connectivity based on open-source software. The proposed lab kit uses Arduino^TM Uno as the microcontroller-board with software freely available.

Thus, the target budget for the proposed kit shown in Figure 4 is around $20; this is because it approximates the cost of a textbook. Moreover, off-the-shelf parts shown in Table 1 are selected to make replacement parts easy to obtain.

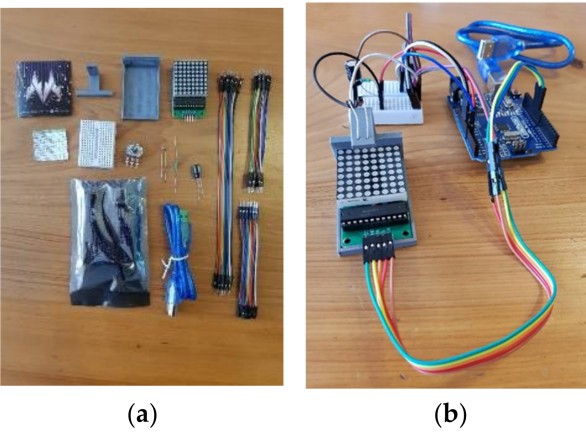

(**a**)  (**b**)

**Figure 4.** Continuous control workstation: (**a**) Required materials; (**b**) An assembled prototype kit.

**Table 1.** Components of a Greenhouse Lab kit.

| Component | Role |
|---|---|
| Arduino^TM UNO | Controller |
| LED Matrix 8 × 8 | Actuator |
| Photoresistor 100 [Ω] | Sensor |
| USB 5 [V] Cable | Connection |
| Potentiometer 10 [kΩ] | Circuit |
| Resistance 1 [kΩ] | Circuit |
| Capacitor 220 [μF] | Circuit |
| Jumpers 10 [cm] | Connection |
| Protoboard | Circuit placement |
| Camping base | Component placement |

Thus, each experiment no longer needs an expensive IO card or similar; instead, the equipment plugs directly fit into the USB port of the computer [12]. Figure 5a shows the experiment design. The students work with a single-input single-output PID feedback structure shown in Figure 4b. Here, the overall light close to the plant is measured by a photoresistor (Sensor). Based on the difference (Error) between the desired luminosity (Set-Point) and the actual lighting conditions (Process Variable), the controller generates the manipulation variable (OP); thus, the number of LEDs on the pane are turned on to achieve the reference.

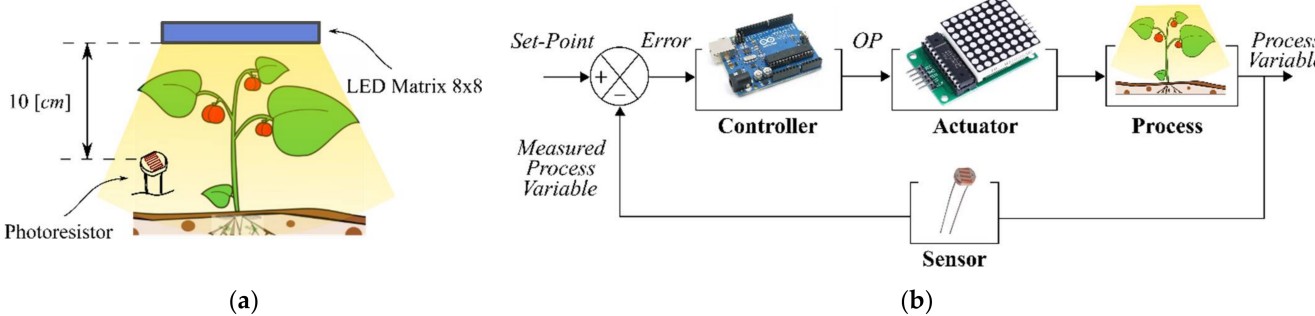

(**a**)      (**b**)

**Figure 5.** Control lab kit: (**a**) Experiment design; (**b**) PID feedback structure.

The present control lab kit could be delivered to students as a connected unit. However, it is strongly recommended that students build the prototypes themselves. It motivates the students as they can look and feel real physical systems even through the COVID-19 lockdowns [42–44]. Additionally, it helps them gain a better understanding of PID feedback structure. The full circuit diagram and schematic diagram is shown in Figure 6.

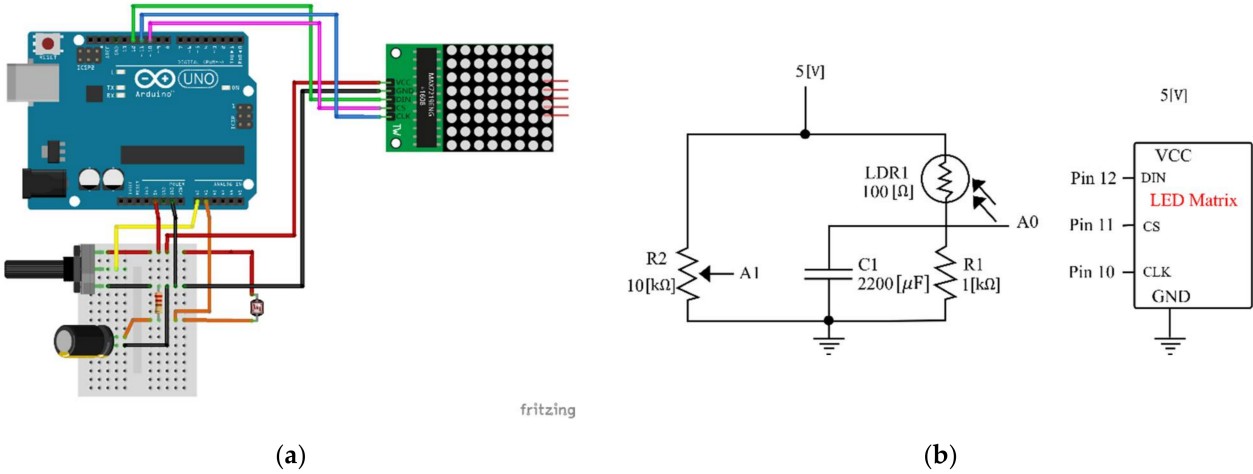

(**a**)      (**b**)

**Figure 6.** Lab kit: (**a**) Circuit diagram; (**b**) Schematic diagram.

As it can be seen in Figure 6, instead of using any design of PCB (printed circuit board) that could increase the cost of the prototype, a protoboard and jumpers are used to connect the components of the kit. Here, the reference is established by a 10 [kΩ] potentiometer. Moreover, located at 10 [cm] in front of the LED Matrix 8 × 8, a 100 [Ω] photoresistor is used as a sensor to measure the lighting conditions. Finally, the charge of a 220 [μF] capacitor is limited by a 1 [kΩ] resistance and monitored by the analog input A1 of the Arduino$^{TM}$ Uno.

### 2.3. Pre-Laboratory Assignment

The primary software package used is Arduino$^{TM}$. However, MATLAB/Simulink$^{TM}$ is also used. Based on this, in a preliminary laboratory session, students installed MATLAB$^{TM}$ version 2016b or later with the Arduino$^{TM}$ support package on their computer. Campus-wide MATLAB/Simulink$^{TM}$ software licenses are available for students and professors at Tecnologico de Monterrey. For institutions which do not have this free subscription, Scilab$^{TM}$ could have been used as an alternative [3].

Although MATLAB/Simulink$^{TM}$ is familiar to the students, in earlier laboratory sessions, students are devoted to up-skilling this language, particularly in:

- basic functions,
- plotting tools,

- edit figures,
- app designer introduction.

The remaining skills (i.e., read data from text file and create animated plots) could be easily taught in a laboratory session. Additionally, based on control engineering knowledge taught in theoretical lessons, students built a system model in Simulink$^{\text{TM}}$ to analyze the main difference between open loop and closed loop response. The purpose of this is to reinforce basic and advanced control theory before working with the real physical device.

### 2.4. Experiment Setup

The experimental setup considers different scenarios, considering the pre-COVID-19, COVID-19 and post-COVID-19 courses. This will demonstrate the capabilities of a portable low-cost kit for the learning experience.

In relation to pre-COVID-19, the course is thought to use traditional materials of a common classroom: the professor presents and explains the content using slides, simulations and develops examples on blackboard. This same experience can be given on the online courses with digital tools, and on the hybrid courses.

In relation to the experiment presented, the same professors from different campuses of Tecnologico de Monterrey have been teaching the control engineering topics on the traditional, online and hybrid courses. The reason the study considers that the professors have been teaching the subject over the different educational models allows us to study the impact of the kit on the course and the perception of the students.

### 2.5. Methodology

In general, once the basics of the control system theory have been taught in theoretical sessions, and assuming that the pre-laboratory assignment has already succeeded, the students must perform the following steps to carry out the Lab-Tec@Home experience (as visualized in Figure 3):

1. Build the prototype.
2. Connect the kit using the open-source software Arduino$^{\text{TM}}$.
3. Apply a step response test and collect real data.
4. Model the process by a transfer function with time delay using three different graphical methods (Ziegler–Nichols, Miller and Analytic method).
5. Use MATLAB$^{\text{TM}}$ to plot the real data and the output mathematical models, Figure 7.
6. Select the best model representation.
7. Design and implement a PID controller using at least two tuning methods (i.e., ZN 1/4 Decay, IAE servo control).
8. Set a reference and collect performance data.
9. Design a human interface using an App Designer to analyze results; then, go to step 7 to adjust the controller.
10. Obtain the performance index and compare them under the two tuning methods.

The open loop step response helps to build the transfer function between the voltages on the capacitor due to lighting conditions. Thus, considering three graphical methods, students obtain the continuous-time mathematical model:

$$G(s) = \frac{K}{\tau s + 1} e^{-\theta s} \tag{1}$$

where $K$ is the gain of the process, $\tau$ stands for the time constant and $\theta$ represents the time delay of the system. Figure 7 presents the real system output and the simulation output using the three different first-order plus time-delay (FOPTD) graphical methods (Ziegler–Nichols, Miller and Analytic method). As it can be seen, there is a good agreement among the two responses obtained from the Miller and the Analytic method, indicating that the model performs well.

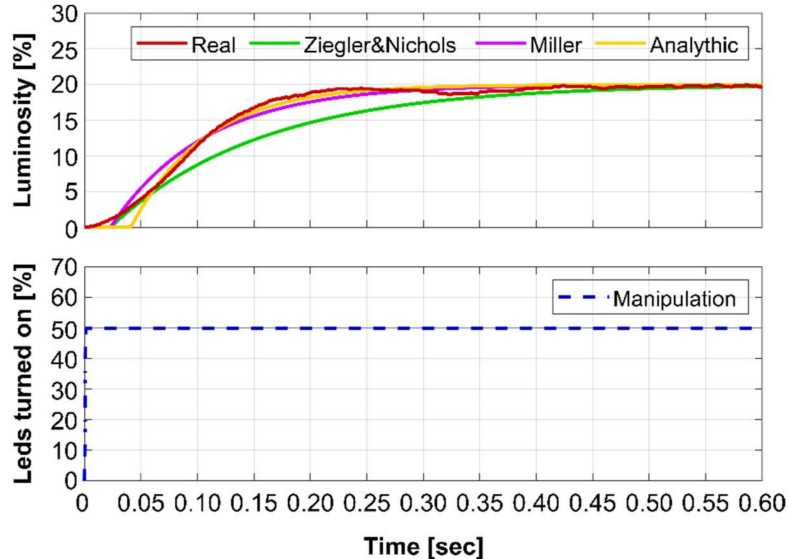

**Figure 7.** Output response coming from the three models.

Furthermore, according to step 7, students use an ideal form to design and implement a PID controller using at least two tuning methods (i.e., ZN 1/4 Decay, IAE servo control). In addition, they designed a human interface using an App Designer to analyse the index performance of the system (i.e., overshoot, peak time, rise time, settling time, decay ratio, etc.) under the control structure. Here, the main learning objective of the course corresponds to understanding the impact of the controller parameters for a desired response.

On the other hand, considering that a graphical programming environment can not completely replace the real-life physical labs, but a combination of both is highly valued in engineering education, students must compare their results using Simulink. See Figure 8.

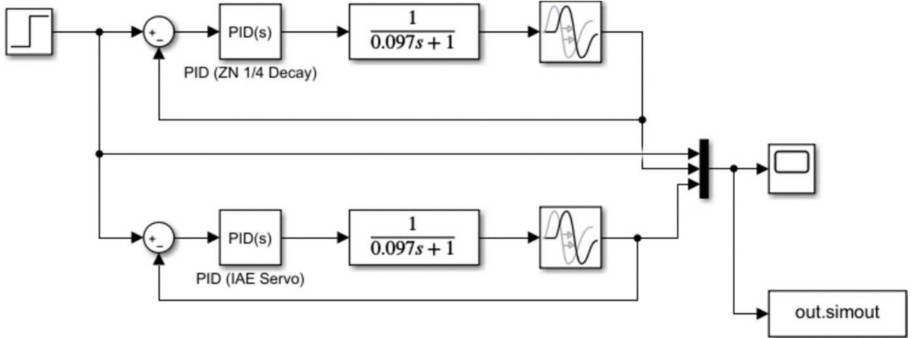

**Figure 8.** Control design using Simulink.

## 3. Related Research Works

Considering that universities and engineering schools are today expecting real solutions for the distance learning, Table 2 presents a detailed comparison with similar kits developed at other universities. However, in previous research works, these kinds of kits have not yet been tested for use in an online only control engineering course.

Table 2 shows the comparison between the proposed kit and previous kits [3,13–16]. The previous prototypes either use many components or do not use open-source software; this implies a higher cost and turns the idea of a totally remote course into a difficult task. However, the proposed kit allows students to carry out experiments at home in their own time.

**Table 2.** Implemented lab kits comparison.

| Lab Kit | No. of Components | Electronics Platform | Open-Source Software | Cost (USD) |
|---|---|---|---|---|
| [3] | 6 | Arduino | Yes | $28 |
| [13] | 8 | MyDaQ | No | $69 |
| [14] | 9 | Arduino | Yes | $416 |
| [15] | 4 | PowerPLC Bridge | No | NA |
| [16] | 15 | Raspberry Pi | Yes | $130 |
| This work | 5 | Arduino | Yes | $20 |

There is a widespread notion that educational systems should empower learners with skills and competencies [45]. In control engineering courses, objectives are specifically stated; thus, they tend to be overly technical [46]. By using the proposed lab kit, the students face experimental errors and possible "failures" which lead them to internalize the problem and troubleshoot it to perform their learning process. The experiential learning (learning-by-doing) promotes different and complex competencies required by ABET Criterion [16,47], such as: (i) critical analysis/thinking, (ii) metacognition, (iii) synthesis of complex problems, (iv) experimental and analytical skills, (v) data collection and processing. Thus, the experimental kit enhances students' engagement and motivation, providing an overall increase in their learning experience and sense of accomplishment and purpose during the course, which is reflected in the professor performance score.

**4. Assessment of Effectiveness**

At each scholar period, the alumni of Tecnologico de Monterrey carried out an anonymous institutional student survey (ECOA) to provide feedback on the course. After the course ends, the professor receives the comments and the final score. The evaluation point is scaled from 1 to 10 with 10 being highly satisfied. Table 3 shows the review criteria:

**Table 3.** Criteria evaluated in an anonymous institutional student survey (ECOA).

| Criteria | Description |
|---|---|
| 1. MET | Regarding the methodology and learning activities. The professor gave me clear and precise topic explanations for me to apply innovative techniques and technological tools that simplified and supported my learning. |
| 2. ASE | About the relationship between the student and professor during the learning process. The professor supported me to resolve doubts; he was available in previously agreed times, and there was a respectful and open learning environment. |
| 3. RET | According to the intellectual challenge level. The course motivated me, and it required me to give my best effort to comply my tasks with quality; this benefitted my learning and my personal growth. |
| 4. APR | Considering the professor as a learning guide. He inspired me and showed commitment to my learning, development, and holistic growth. |
| 5. REC | Would you recommend a friend to take classes with this professor? |
| 6. MEJ | Do you consider the professor as one of the best teachers you have ever had? |
| 7. PRA | Related to the understanding of concepts in terms of their application in practice. I solved real cases, projects, or problems. Moreover, I conducted practices in laboratories or workshops, visited companies or organizations, and interacted with people who work applying the topics. |
| 8. IMF | How is the interaction with my professor in the MFD class during the COVID-19 pandemic? |

Having considered the results of the ECOA, it is apparent that teaching the MFD course is one of the major concerns of professors who have been well evaluated for years under the classical classroom-based learning model. More specifically, it represents a big challenge for professors who teach courses strongly related to laboratories (i.e., control engineering courses).

Based on Table 4, the following subsections present the average results of the ECOA survey trialed on 290 students who attended the course from different campuses in Mexico. These results are compared to scores obtained from 165 students who attended the course in previous scholar periods, where just simulation and virtual labs were carried out.

**Table 4.** Student impact of using lab kits.

| | Without Kit | | | | With Kit | | |
|---|---|---|---|---|---|---|---|
| | **Characteristics** | | | | **Characteristics** | | |
| **Period** | **Students** | **Attended Campus** | **Learning Model** | **Period** | **Students** | **Attended Campus** | **Learning Model** |
| Feb–Jun 2019 | 31 | 1 | Classroom | Jul–Aug 2020 | 32 | 8 | MFD |
| Jul–Aug 2019 | 41 | 1 | Classroom | Aug–Dec 2020 | 116 | 2 | MFD |
| Aug–Dec 2019 | 36 | 1 | Classroom | Jan–Feb 2021 | 53 | 4 | MFD |
| Jan–Feb 2020 | 32 | 1 | Classroom | Feb–Jun 2021 | 64 | 5 | MFD |
| Feb–Jun 2020 | 25 | 1 | MFD | Jul–Aug 2021 | 25 | 5 | MFD |

### 4.1. MET Criterion Results

Figure 9 shows the professor scores per scholar period according to criterion 1. MET, in which students evaluate the methodology and learning activities. MFD courses started in early March 2020; here, the lab kit is not implemented, and the professor obtained the lowest score. On the other hand, summer 2020 is the first period in which the lab kit is implemented; thus, the student survey reveals better, and more constant results compared to previous scholar periods. Furthermore, from the winter scholar period 2021 onward, Tecnologico de Monterrey presents other learning models that integrate innovative teaching strategies (i.e., synchronous remote face-to-face hybrid, HPRS, and hybrid face-to-face duplicate plus activities, HPDA); here, the lab kit could also be well implemented.

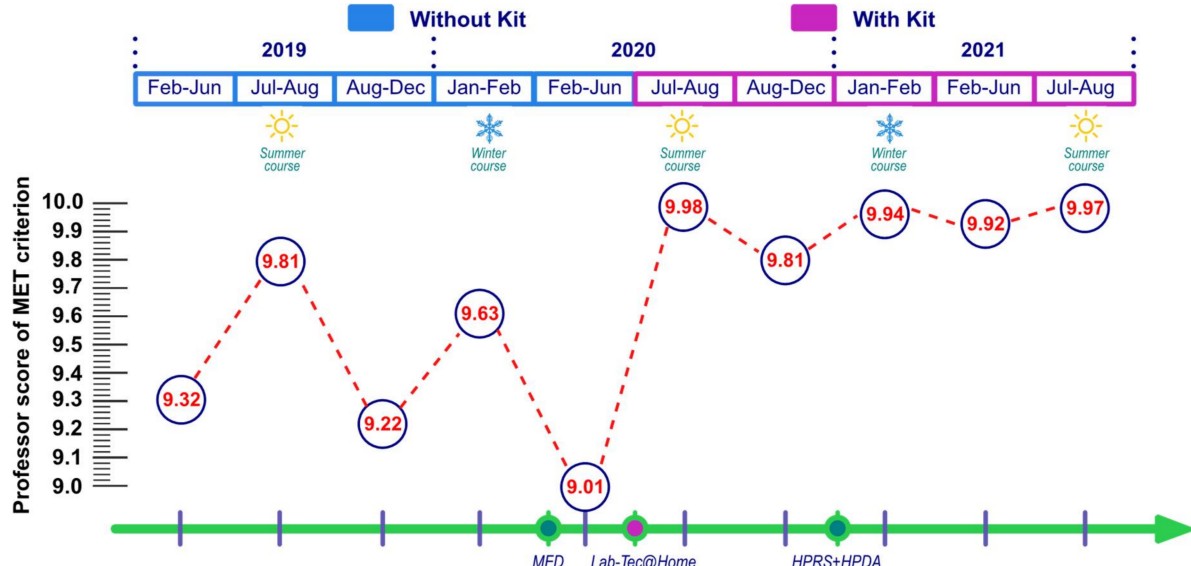

**Figure 9.** MET results.

### 4.2. ASE-RET-APR-REC-MEJ Criteria Results

The radar plot in Figure 10 shows a comparative analysis of the professor's performance in an online course versus a conventional face-to-face class. The criterion 2. ASE shows an overall final score of 9.898, which means that students appreciated the influence of the professor during the learning process (i.e., proximity, feedback, open learning environment, etc.). Moreover, in regards to the criteria that evaluate the professor's effectiveness,

excellent results are achieved: 4. APR, 9.925, and 5. REC, 9.954, out of 10. Thus, the learners highly recommend that students must take classes with the professor. (See Appendix B, I); this is despite the limitations and concerns of online courses and workshops at a global scale due to the COVID-19 pandemic [48,49].

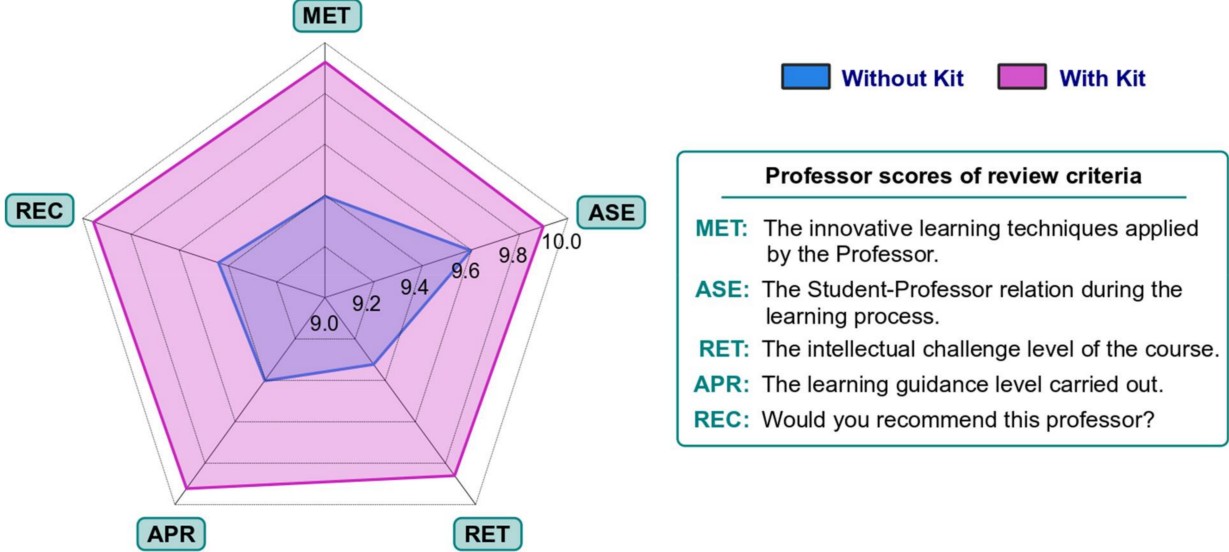

**Figure 10.** ASE-RET-APR-REC-MEJ results.

In addition, students perceived that the professor improved the course using the lab kit. Here, learners picked up practical skills which made them feel motivated and inspired to carry out the tasks with their best effort [50] (See Appendix B, II). Thus, after carrying out experimental setups and related tools, student expressed that they faced a high intellectual challenge level course (See Appendix B, III). This can be appreciated in the final score of criterion 3. RET, 9.862. Finally, the average score of the professor, according to criterion 1. MET, described in the previous subsection, is also presented in the radar plot; this visualizes the overall performance improvement scores of the professor applying this innovative technique.

The professor performance comparison shown in Figure 10 agrees with the outstanding score obtained in the 6. MEJ criterion; it rates 0.894 out of 1. Here, students strongly agreed that the professor is one of the best teachers that they have ever had. (See Appendix B, I)

### 4.3. PRA Criterion Results

Although education has been forced to change to distance learning [49], students must receive the same level of educational quality that they receive in a typical classroom-based course structure [51]. Moreover, in courses which involve laboratory sessions, it is crucial for the professor that the students understand the concepts in terms of their application in practice; this is evaluated in criterion 7. PRA. The comparative results are shown in dispersion plot, Figure 11. As it can be seen, by using the kit, the overall final score of the professor is improved from 9.1176 to 9.7655. Moreover, even though the courses are offered to more than one campus, the standard deviations of the responses are outstanding and are reduced by more than half; they passed from 1.561 to 0.6.

Students agreed that by carrying out the workshop and applying the topics of the class, they obtained a tangible experience in control system design; thus, it helped them to understand how the feedback control structure works in a real process. (See Appendix B, IV).

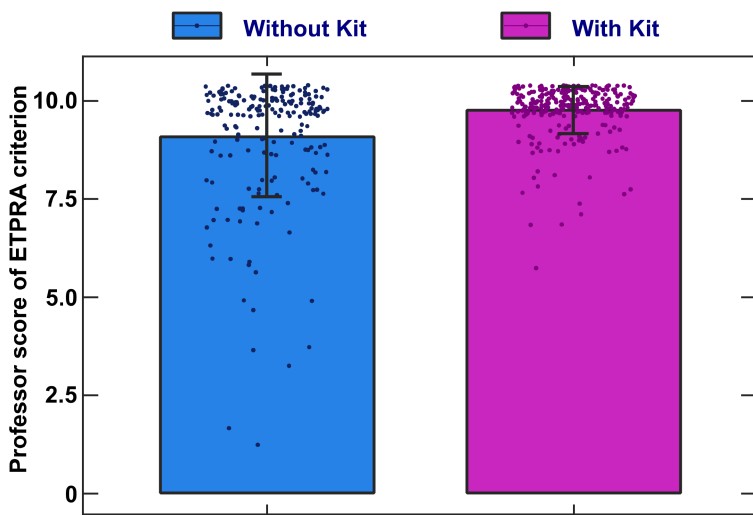

**Figure 11.** PRA results with and without the lab kits.

### 4.4. IMF Criterion Results

To circumvent the crisis of the COVID-19 pandemic, Tecnologico de Monterrey moved the courses into a virtual environment. Here, e-learning resources, attractive lectures, well-planned course time, particular rules and policies, innovative teaching strategies, and a closer learning guide play key roles in achieving a rich online learning experience. In addition, it encourages the student–professor interaction; it is evaluated in criterion 8. IMF. The lowest criterion score was 8.58, which was obtained during the Feb–Jun 2020 scholar period; here, the lab kit was not implemented. However, once the lab kit was implemented, the criterion score improved to 9.559 with a standard deviation of 1.19. Thus, the outstanding results are shown in Figure 12 by a Pareto chart. Here, 223 students out of 290 (more than the 90 percent of the students) expressed that even though the massive course is taught under the MFD learning education model, there exists an excellent interaction with the professor (See Appendix B, V). Henceforth, school heads shared these results to motivate teachers for their growth and development under remote courses.

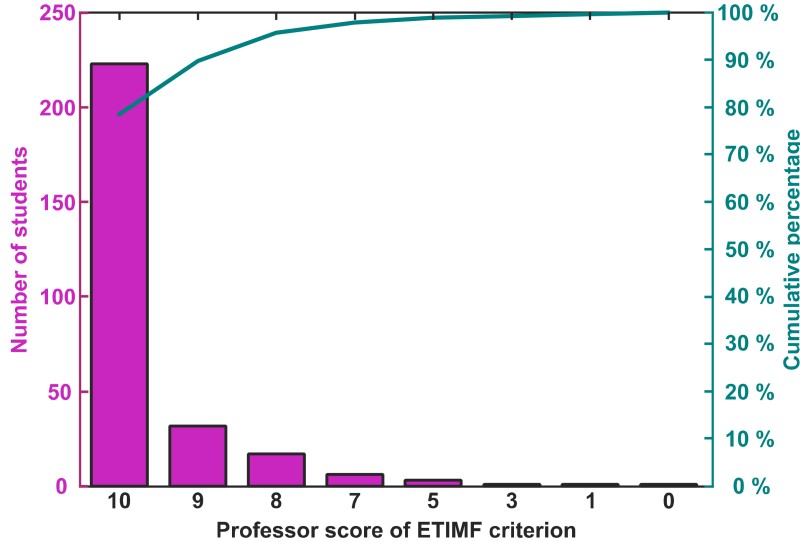

**Figure 12.** IMF results.

### 4.5. ECOA Findings

As it can be seen in the optional institutional survey, the students rate the kit as a useful tool to encourage them to learn about control engineering topics. In fact, chemical

engineers with minimal electronic circuit and MATLAB$^{TM}$ competencies found this project attractive and challenging. Thus, during the course the students show:

- engagement,
- intrinsic motivation,
- inspiration in teaching-learning activities,
- connection between practice and theory.

Henceforth, at the end of the course, the students showed a high confidence in control engineering topics. Thus, this educational project can be adopted by others when lab kits could be designed for conventional control strategies such as cascade, feedforward, and ratio control in mechanical and chemical processes [52–54]. In addition, lab kits could be designed for graduate students to teach nonlinear control [55] (i.e., flexible joint mechanism, inverted pendulum, etc.) and model predictive control [56] (i.e., tape transport and jet aircraft system).

### 5. Assessment of Students' Performance on the Courses

The student performance is commonly measured by the final score obtained on the course. As considered on the experiment setup, the same professor teaches the course and the only key difference is the kit. The grading scale at Tecnologico de Monterrey is set from 100 as the best score to 0 as the worst score. In Figure 13, a result comparing the average score obtained by the students is presented; a 16% difference is presented in the groups which implemented the kit on the courses. Moreover, by making this assumption, it could be concluded that the application of the theoretical concepts on practical cases increases the students' knowledge and can be reflected on the final scores.

**Figure 13.** Score average comparison between cases with the kit and without the kit.

In Figure 14, the analysis of the score distribution in 10 point ranges is presented. Notice that the results of both studies, with the kit and without the kit, present a similitude with a normal distribution curve. On one hand, it can be observed that in the case without the kit, the centered value is in the score range from 70 to 79 with 38.66% of students under this score. On the other hand, it can be observed that in the case with the kit, the centered value is in the score range from 100 to 90 with 46.53% of the students under this score. As shown in Figure 14, the tendency that shows the grades from the students with the kit change the center value to a higher score because of the introduction of the kit to reinforce the learning experience.

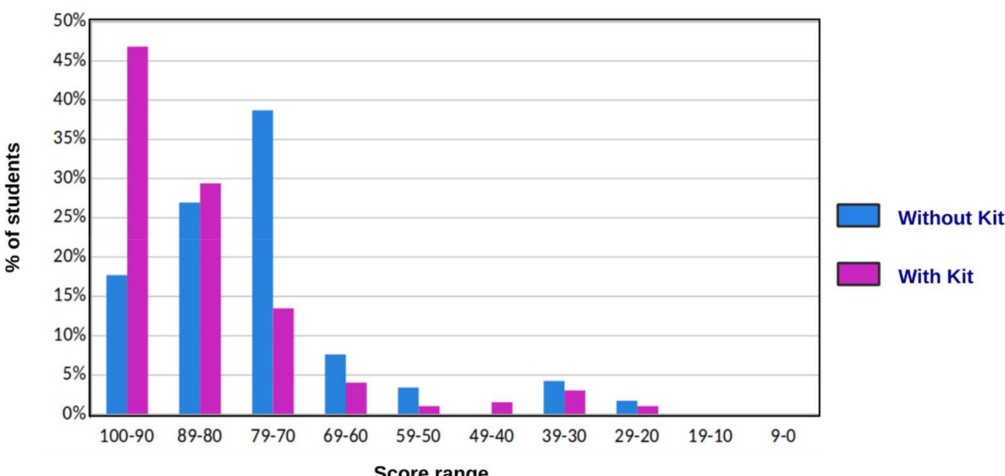

**Figure 14.** Score distributions between cases with the kit and without the kit.

Students fail the course with a score less than 70. In Figure 14, it can be observed that the percentage of students that fail the course without the kit is 16.81%, and the introduction of the kit changes the failing to 10.89%. Therefore, it could be observed that the kit reinforces the students' learning and reduces the failing percentage.

The results presented in Section 4, assessment of effectiveness, and in this section, present solid findings that implementing the Lab-Tec@Home has a significant effect on the student comprehension of a difficult subject as automatic control, and is related to the overall evaluation of the course and the professor. The implementation of a kit, as presented in this paper, can reinforce the knowledge of the student as demonstrated by the statistical data from the average of the final scores. In addition, the difficult and abstract concepts that involve control engineering can be better explained using a simple kit to demonstrate the effects that are present; if the teacher gives a practical example, the students can relate to the concepts applied in the class with relative ease.

## 6. Conclusions

The present work describes the educational project Lab-Tec@Home carried out for massive flexible digital courses at Tecnologico de Monterrey, Mexico. Here, based on an inexpensive lab kit for control engineering education and using an entire open-source software Arduino$^{TM}$, learners assemble the kit and plug it in their own computing devices at home. It motivates students as they can look and feel real physical systems even through the COVID-19 lockdown. Furthermore, they can gain the benefits of working on real experiments in their own time, and are not constrained by the short access times that occur in a standard laboratory. Moreover, this project overcomes the lack of network capacity commonly present for large size classes or remotely/rurally located students who attend virtual and remote labs. Although the courses are taught online, more than 290 students, enrolled in different campuses, showed a high level of confidence in controlling engineering topics. They also achieved a superior learning outcome, and found this educational project attractive and challenging. This is expressed in the institutional student survey, ECOA, where outstanding results are obtained. Thus, the present research work is based on an innovative and easy-to-use tool kit for designing take-home lab exercises. This is useful for teaching PID controller design remotely at the undergraduate level. It enhances the learning experience, and it can be extended to teach advanced control strategies.

**Author Contributions:** Conceptualization, C.S. and D.S.; methodology, C.S., D.S. and R.A.R.-M.; validation, E.A.L.-G. and D.N.-D.; investigation, C.S., D.S. and A.V.-M.; resources, D.N.-D. and E.N.-J.; writing—original draft, C.S. and D.S.; writing—review and editing, R.A.R.-M. and A.V.-M.; supervision, C.S. and D.S.; project administration, D.N.-D., E.N.-J. and A.V.-M.; funding acquisition, R.A.R.-M. and A.V.-M. All authors have read and agreed to the published version of the manuscript.

**Funding:** The authors would like to acknowledge the financial and the technical support of Vicer-rectoria Academica y de Innovacion Educativa, of the Tecnologico de Monterrey, in the production of this work, especially the scientometrics team led by Nathalíe María Galeano Sánchez, and to the Faculty Development and Innovative Education Center for the pedagogic support during this work. The current project was funded by Tecnologico de Monterrey and Fundación FEMSA (Grant No. 0020206BB3, CAMPUSCITY Project). Additionally, the authors would like to acknowledge the financial support of NOVUS 2020- 140-, an initiative of Tecnologico de Monterrey, Mexico, in the production of this work.

**Data Availability Statement:** Data is contained within the article.

**Acknowledgments:** The authors also thank student Juan Carlos Triana Vela for his fruitful discussions, figures and tables editing, and especially for the analysis of the data.

**Conflicts of Interest:** The author declares no conflict of interest.

## Appendix A

Name: Proposal

Light-emitting diodes (LEDs) are potentially one of the biggest advancements in horticultural lighting in decades. LEDs can play a variety of roles in horticultural lighting, including use in controlled environment research, lighting for tissue culture, and supplemental and photoperiod lighting for greenhouses. Nowadays, recent studies have been carried out to analyze the influence of supplementary LED lighting in modern tomato production. In these studies, tomato plants (cv. Lyterno) were grown in a commercial-like greenhouses and exposed to different lighting conditions [28,29]. Results revealed that supplementary LEDs induced the accumulation of secondary metabolites in tomato leaves such as higher rutin concentrations in young leaves and partly in mature leaves [30–33], which can be used for future research works in: medicines, antibiotics, insecticides, herbicides, biomass generation, etc. [34]

Based on this, you have to design and implement a control system to regulate the lighting conditions in a greenhouse; thus, using the prototype kit.

## Appendix B

Name: Free-text responses from the institutional student survey (ECOA)

I.  Here, the learners highly recommend taking classes with the professor, and they strongly agreed that the professor is one of the best teachers that they have ever had. Thus, the following main responses are encountered:

- *"Highly recommended because the three professors tried to make the best use of the situation. They have coordinated very well, they complement each other stupendously, they are very cool with the students, and they are always free and accessible without any doubt about the class. There is a great deal of merit that they made in research looking for a KIT to help us in the understanding of the subject and to take this practical part of it. I did not expect a lot from the hybrid model but I'm very satisfied with the result, congratulations."*

- *"The classes went nicely; the professor gives you the confidence to ask and solve your doubts with very clear explanations. 100% recommended. One of the best professors I had."*

- *"Both of you are the best teachers I have had in my whole career; you are very dedicated and committed. Thank you very much by giving this time for my preparation."*

II. The learners picked up practical skills which made them feel motivated and inspired to give their best effort to carry out the tasks. Hence, the following response are:

- *"He and his brother are some of the most inspiring professors I ever had; they were very kind and were available to help their students. In addition, their classes were challenging but always gave us the weapons needed to learn and understand the class material. And because of that, I feel prepared for real life situations. Love you!"*
- *"The professor was very attentive, always solving doubts and answering comments, giving some time for teamwork-based learning and always shown to be accessible. Additionally, he had a good mood while giving the class which motivates the students."*
- *"Inspiring and attentive to the needs of the students. Excellent professor in all the ways possible. One of the best professors I ever had."*

III. The students expressed that they faced a high intellectual challenge level course. Then, the main responses are:

- *"One of the best professors I've ever had. He really takes care of the intellectual development of each of the students; it is visible that he prepares all of his classes in order so every one of us can understand the class content in a better way, as well as interactively. It also inspires you to deepen your learning on the subject topics."*
- *"The best teachers in the world. They give very good explanations. The homework was challenging. They give you academic counselling when you need it and they have a lot of patience."*
- *"He gives you very clear explanations, showing the mathematical proof of each topic and procedure, and then solving exercises where you can apply the knowledges which makes learning a lot easier. The homework is more complex than class exercises, but not impossible, and that, in my opinion, is helpful for studying and understanding each of the topics. He is so attentive with all his students, and very patient when solving doubts."*

IV. Students agreed that by carrying out the workshop and applying the topics of the class, they obtained a tangible experience in control system design. Thus, the following main responses are encountered:

- *"This was the first time I took a subject with three teachers and it was one of the best experiences of Tecnologico de Monterrey. It'll be a privilege to take another class with them. They really know about what they are teaching and share their knowledge with you, and also have exceptional patience. They look for you to understand the whole topic and that you are ready to keep going. They teach both theorical and practice parts of the subject. You can clearly see how the theory is applied in real life. You are amazing doctors, never change! Keep inspiring more generations. I'm really inspired because of you. Thank you for being my teachers!"*
- *"The class was dynamic and easy to understand. It is obvious that professors do take care of us to understand the class topics and how to apply them by solving problems. I would like to take the subject of Process Automatization Laboratory with you."*

V. Students expressed that even though the massive course is taught under the MFD learning education model, there exists an excellent interaction with the professor. Thus, the main response are:

- *"Excellent professor. He is dedicated to make you learn and to do your best, despite the online classes situation. In addition, he is always available to help you with your doubts and make sure that you understand 100%. They make a very enjoyable class."*
- *"One of the best university experiences I've ever had. They are kind and charismatic, and they know a lot about the subject. Their classes are up to date, and they have a lot of experience. It is quite hard that teachers give you personal attention during the pandemic, and even more so in class. Both professors achieved making their students feel like that they were included, that they learned something and that they participated. The best learning experience I've had in university; thank you very much!"*

- *"The Drs. Sotelo class was quite dynamic and they take care that each of their students have understood the explanations during the course. Extra points awarded because the whole subject was online."*
- *"I really liked the breakout rooms methodology. Comparing it with my classes in Germany, these spaces give you a lot more freedom to make questions without the fear of interrupting the class. It helps a lot that homework is being done inside break out rooms, because that's how I have the opportunity of asking my partnerstoo and to see other solutions."*

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
