# Peer review of "Lab-Tec@Home: A Cost-Effective Kit for Online Control Engineering Education"

_electronics, doi:10.3390/electronics11060907_

Round 1
Reviewer 1 Report
Dear authors!
The work presents an elegant solution to online engineering education problems using low-cost Arduino-based kits. The high relevance of this study is conditioned by both trends in online education and the pandemic.
The proposed kit helps students to understand such topics as PID-control and implement a real control process themselves. It is shown that using the kit helps the students to learn the topic more successfully. While the work has been performed at a high level, several remarks should be made.
1. Figure 4 depicts the plant in the feedback loop while the designed controller takes a signal from the overall light but not the plant. Probably, a greenhouse should be depicted instead. The question arises: is it intended that LED compensates the reducing sunlight in the twilights, or for other reasons (e.g. bottom leaves are in shadow due to the plant growth)?
Or a drawing in Fig. 1 should clarify where the LED pane and the photoresistor are placed in a proposed setup: under the plant, or near it, and at what level. Please make the experiment design clear.
2. The Introduction covers mostly educational projects but does not answer the question of why Arduino is the most feasible solution and what capabilities it has. To answer this question, I suppose, several projects implemented in this type of hardware should be mentioned, including educational, students' projects, and industrial/scientific prototypes.
For example, there have been recently published works on a Lightsaber for student education, a data acquisition system for beehive, a number of autonomous robots, etc. based on Arduino. Even some rather serious Arduino-based projects have been reported, such as an Arduino-controlled anaerobic bioreactor and a painting robot with advanced tone rendition technique. Please include a brief literature overview on the topic and add some relevant literature sources to the reference list.
My overall impression is very good, and I suppose, the paper can be published after a minor revision.
Author Response
Dear Reviewer 1, please find below our detailed responses to your comments and suggestions. Our response is always in italic. Changes in revised manuscript are highlighted in yellow in the attached document.
Reviewer 1:
Dear authors!
The work presents an elegant solution to online engineering education problems using low-cost Arduino-based kits. The high relevance of this study is conditioned by both trends in online education and the pandemic.
The proposed kit helps students to understand such topics as PID-control and implement a real control process themselves. It is shown that using the kit helps the students to learn the topic more successfully. While the work has been performed at a high level, several remarks should be made.
Authors do appreciate your consideration and useful comments on the manuscript. It surely improves the quality of the paper. Based on these valuable comments, the article has been revised and the modified version is presented.
- First comment
Figure 4 depicts the plant in the feedback loop while the designed controller takes a signal from the overall light but not the plant. Probably, a greenhouse should be depicted instead. The question arises: is it intended that LED compensates the reducing sunlight in the twilights, or for other reasons (e.g. bottom leaves are in shadow due to the plant growth)?
Or a drawing in Fig. 1 should clarify where the LED pane and the photoresistor are placed in a proposed setup: under the plant, or near it, and at what level. Please make the experiment design clear.
Thank you for your observation. We apologize for any misunderstanding. The controller takes the difference between the desired luminosity and the overall light close to the plant, then, the number of leds of the pane are turned on to achieve the reference. The goal of our proposed kit is not to faithfully reconstruct a specific real greenhouse, but to set up a replica of a greenhouse for tomato plants in which students can identify the components and variables involved in a feedback control loop. Thus, the students can apply the contents of the class, such as: system model identification, design of control strategies, computation of performance indexes for control systems, etc.
Based on this, lines 118-120 (Section 2, page 3) were added for the revised manuscript. Additionally, Fig. 4 (Section 2, page 5, in the previous manuscript) and lines 139-142 (Section 2, page 4, in the previous manuscript) have been modified for the revised manuscript.
Furthermore, to clarify the experiment design, lines 153-156 (Section 2, page 5 in the previous manuscript) were modified for the revised manuscript. Thank you.
- Second comment
The Introduction covers mostly educational projects but does not answer the question of why Arduino is the most feasible solution and what capabilities it has. To answer this question, I suppose, several projects implemented in this type of hardware should be mentioned, including educational, students' projects, and industrial/scientific prototypes.
For example, there have been recently published works on a Lightsaber for student education, a data acquisition system for beehive, a number of autonomous robots, etc. based on Arduino. Even some rather serious Arduino-based projects have been reported, such as an Arduino-controlled anaerobic bioreactor and a painting robot with advanced tone rendition technique. Please include a brief literature overview on the topic and add some relevant literature sources to the reference list.
Authors appreciate Reviewer’s comment, thank you. As it is well mention in [1], when lab kits are used for control engineering education, the major drawback is that they are quite expensive, and thus not affordable by many educational institutions. Here, the development of affordable control kits which can be purchased and used by everybody (students and teachers) is a highly desirable feature [2]. For this reason, in [3][4][5][6][7] Arduino is used as a platform to develop low cost lab kits to test and validate engineering theoretical concepts. Similarly, in [1][8][9][10], it is used to handle the problem of the real-time data recording and perform feedback control experiments.
Three factors distinguish Arduino: i) it is small size, portable and ready to be used; ii) the software is freely available both for Python and MATLAB/Simulink; iii) supporting educational videos and material are freely made available. Thus, a lab kit platform based on an Arduino microcontroller provides a complete, easy-to-use hardware and software solution for designing take-home lab exercises [3][11][12].
On the other hand, based on the Reviewer’s suggestion, research works [1][2][3][4][5][6] [7][8][9][10][11][12], listed at the end of this response, and some sentences in Section 1 (page 2, lines 62-64, lines 66-68, and lines 70-75) were added for the revised manuscript.
My overall impression is very good, and I suppose, the paper can be published after a minor revision.
Authors do appreciate the Reviewer comments. Thank you.
We believe that these corrections and additions adequately address all the points that have been raised by the Reviewer. Authors also believe that the comments significantly helped to improve the manuscript.
We hope that the paper can be accepted for publication.
On behalf of all authors,
Carlos Sotelo, Ph.D.
Professor
Mechatronics and Electrical Department,
Sciences and Engineering School,
Tecnologico de Monterrey, Monterrey Campus
64849 Monterrey Nuevo León, México.
Email: carlos.sotelo@tec.mx
References:
[1] de Moura-Oliveira, P. B; Hedengren, J. D.; Rossiter, J. A. Introducing Digital Controllers to Undergraduate Students using the TCLab Arduino Kit. IFAC-PapersOnLine. 2020, 53(2), 17524-17529.
[2] Rossiter, J. A.; Pope, S. A.; Jones, B. L.; Hedengren, J. D. Evaluation and demonstration of take home laboratory kit. IFAC-PapersOnLine. 2019, 52(9), 56-61. https://doi.org/10.1016/j.ifacol.2019.08.124.
[3] Sarik, J.; Kymissis, I. Lab kits using the Arduino prototyping platform. Proceedings - Frontiers in Education Conference (FIE), Arlington, VA, USA, 27-30 Oct. 2010; IEEE: Manhattan, New York, USA.
[4] Alvarado, I.; Maestre, J. M. A Lightsaber to Introduce Students to Microcontrollers. IFAC-PapersOnLine. 2019. 52(9), 139-143.
[5] Oltean, S. E. Mobile robot platform with Arduino uno and raspberry pi for autonomous navigation. Procedia Manuf. 2019. 32, 572-577.
[6] KüçükaÄŸa, Y.; Facchin, A.; Torri, C.; Kara, S. An original Arduino-controlled anaerobic bioreactor packed with biochar as a porous filter media. MethodsX. 2022, 101615.
[7] Prima, E. C.; Karim, S.; Utari, S.; Ramdani, R.; Putri, E. R. R.; Darmawati, S. M. Heat transfer lab kit using temperature sensor based arduinoTM for educational purpose. Procedia Eng. 2017, 170, 536-540.
[8] Irigoyen, E.; Larzabal, E.; Priego, R. Low-cost platforms used in Control Education: An educational case study. IFAC Proceedings Volumes. 2013, 46(17), 256-261.
[9] Reguera, P.; García, D.; Domínguez, M.; Prada, M. A.; Alonso, S. A low-cost open source hardware in control education. case study: Arduino-feedback ms-150. IFAC-PapersOnLine. 2015, 48(29), 117-122.
[10] Ebrahim, A. Lab kit for liquid level measurement using valve opening with PID controller: Ahmed Abdulhameed Ebrahim, Yvette Shaan-Li Susiapan. TSSA. 2020, 3.
[11] Oteri, O. M. The arduino e-kit as applied in engineering, science and technology e-learning. 2020 Sixth International Conference on e-Learning (econf), Sakheer, Bahrain , 6-7 Dec. 2020; IEEE: Manhattan, New York, USA.
[12] Oliver, J. P.; Haim, F. Lab at home: Hardware kits for a digital design lab. IEEE Trans. Educ. 2008 52(1), 46-51.

Reviewer 2 Report
Thank you for the opportunity to review this manuscript.
It is not clear the aim of your research.
I recommend to delete the sentence, it is redundant information - ”This paper is structured as follows: Section 2 describes the materials and methods used to implement the lab kit under the MFD learning model. In Section 3, a rigorously comparison to related research works is presented. In Section 4, the assessment effectiveness of the educational project is analyzed. Finally, conclusions and future research works are presented in Section 5.”
I recommend to add at Materials and Methods section a few new subsections: Participant, Design and procedure of research.
I recommend to highlight the strength points and the limitations of your research.
Author Response
Dear Reviewer 2, please find below our detailed responses to your comments and suggestions. Our response is always in italic. Changes in revised manuscript are highlighted in yellow in the attached document.
Reviewer 2:
Thank you for the opportunity to review this manuscript.
- First comment
It is not clear the aim of your research.
Thank you for your comment. Authors address the circumvent drawback of using lab kits for control engineering education mentioned in [1][ 2], the development of affordable control kits which can be purchased and used by everybody (students and teachers). For this reason, in the present work, using open hardware and software, the aim of this research is to expand the access of hands-on control education at the undergraduate level during a critical COVID-19 pandemic situation. A kit is assembled for around $20 which is used to cover the functionality of a typical station in an introductory control laboratory. Moreover, as in in [3][4][5][6][7] Arduino is used as a platform to develop the low cost lab kit to test and validate engineering theoretical concepts. Similarly, as in [1][8][9][10], it is used to handle the problem of the real-time data recording and perform feedback control experiments.
While there exists lab kit options for teaching control engineering education [1][8][9][10], the proposed kit replaces expensive equipment with an affordable alternative that can be easily shipped anywhere in the world and used by students who do not have a formal laboratory space to use [1][11][12].
Taking into account your comment, and to highlight the importance and novelty of this research work, some sentences in (page 1, lines 17-18 and page 2 lines 62-64), and references [1][2][3][4][5][6][7][8][9][10][11][12] were added for the revised manuscript.
- Second comment
I recommend to delete the sentence, it is redundant information - ”This paper is structured as follows: Section 2 describes the materials and methods used to implement the lab kit under the MFD learning model. In Section 3, a rigorously comparison to related research works is presented. In Section 4, the assessment effectiveness of the educational project is analyzed. Finally, conclusions and future research works are presented in Section 5.”.
Thank you for your observation. As the Reviewer suggest lines 83-87 (Section 1, page 2, in the previous manuscript) were removed for the revised manuscript.
- Third comment
I recommend to add at Materials and Methods section a few new subsections: Participant, Design and procedure of research.
We thank the Reviewer for this suggestion. We have added a Figure in Subsection 2.1 (page 4) that encompasses the methodology followed by the students, the pedagogical approach followed by the professor. Finally, this methodology is categorized under the Challenge-Based Learning pedagogic technique. The authors believe that this figure could clarify the methodological approach followed.
Regarding the Participants, the course was offered to Chemical Engineering students in order to assess the easiness of developing the kit without previous experience with electrical components as well as to facilitate the meta-cognition process in the students by introducing concepts and hands on activities to relate and practice those concepts under a flipped-classroom environment. The course was offered nation-wide from Summer 2020 to August-December 2021.
Subsection 2.1 was created (page 4, lines 130-151) as suggested by the reviewer in which the Overall Course Methodology is discussed with the above information.
- Fourth comment
I recommend to highlight the strength points and the limitations of your research.
The Reviewer raises an interesting concern, thank you for your observation. Prior the designing of the proposed educational project a research has been done on different types of lab kit used for engineering education [1][3][4][5][6][7][8][9][10]. However, authors found that some of the research works have limitations. Thus, in order to highlight the strength points of the proposed research work, authors present the following main contributions:
- Easy-to-use hardware/software solution for designing take-home lab exercises. To encourage the widespread adoption of distributed lab kits, the proposed lab kit is small size, portable, and ready to be used. Moreover, the software to use the kit is freely available both for Python and MATLAB/Simulink, and supporting educational videos and material are freely made available. Thus, the kit replaces expensive equipment with an affordable alternative that can be easily shipped anywhere in the world and used by students who do not have a formal laboratory space to use [11][12].
- Useful for teaching PID controller design at the undergraduate level. Considering PID type controllers have been commonly used in industrial applications, this lecture topic has a high relevance in most introductory control courses. However, students facing lack of knowledge in controlling the PID as a hardware where this is due to the complexity of the PID controller to be implemented [10]. Thus, using open hardware/software and based on a real situation, the proposed system provides an effective and attractive solution for teaching PID controller design at the undergraduate level.
- Appropriate to be implemented in remote course. Unlike previous research works, where lab kits are used in classroom, in the present work the kit is implemented for massive remote courses. In addition, based on an institutional student survey (ECOA), the results were quite positive where most of the students indicated that although the class was taught totally remote, they pick up practical skills which makes them feel motivated and inspired to give the best effort to carried out the tasks.
Thus, the proposed lab kit delivers a consistent, high quality laboratory experience for students whose participate in distance education.
On the other hand, as previous research works [1][8][9][10] used for control engineering education, the main limitation of the proposed research work is that the lab kit is useful for linear time invariant and single input – single output systems. For nonlinear and multivariable processes mentioned in [42][45] (in the previous manuscript), new lab kits will be carefully designed to teach advanced control strategies such as nonlinear and model predictive control.
Taking into account your comment, and to highlight the strength points and mention the limitation of this research work, some sentences in (page 13, lines 379-385, and page 15, lines 434-437), and references [1][3][4][5][6][7][8][9][10][11][12] were added for the revised manuscript.
We believe that these corrections and additions adequately address all the points that have been raised by the Reviewers. Authors also believe that the comments significantly helped to improve the manuscript.
We hope that the paper can be accepted for publication.
On behalf of all authors,
Carlos Sotelo, Ph.D.
Professor
Mechatronics and Electrical Department,
Sciences and Engineering School,
Tecnologico de Monterrey, Monterrey Campus
64849 Monterrey Nuevo León, México.
Email: carlos.sotelo@tec.mx
References:
[1] de Moura-Oliveira, P. B; Hedengren, J. D.; Rossiter, J. A. Introducing Digital Controllers to Undergraduate Students using the TCLab Arduino Kit. IFAC-PapersOnLine. 2020, 53(2), 17524-17529.
[2] Rossiter, J. A.; Pope, S. A.; Jones, B. L.; Hedengren, J. D. Evaluation and demonstration of take home laboratory kit. IFAC-PapersOnLine. 2019, 52(9), 56-61. https://doi.org/10.1016/j.ifacol.2019.08.124.
[3] Sarik, J.; Kymissis, I. Lab kits using the Arduino prototyping platform. Proceedings - Frontiers in Education Conference (FIE), Arlington, VA, USA, 27-30 Oct. 2010; IEEE: Manhattan, New York, USA.
[4] Alvarado, I.; Maestre, J. M. A Lightsaber to Introduce Students to Microcontrollers. IFAC-PapersOnLine. 2019. 52(9), 139-143.
[5] Oltean, S. E. Mobile robot platform with Arduino uno and raspberry pi for autonomous navigation. Procedia Manuf. 2019. 32, 572-577.
[6] KüçükaÄŸa, Y.; Facchin, A.; Torri, C.; Kara, S. An original Arduino-controlled anaerobic bioreactor packed with biochar as a porous filter media. MethodsX. 2022, 101615.
[7] Prima, E. C.; Karim, S.; Utari, S.; Ramdani, R.; Putri, E. R. R.; Darmawati, S. M. Heat transfer lab kit using temperature sensor based arduinoTM for educational purpose. Procedia Eng. 2017, 170, 536-540.
[8] Irigoyen, E.; Larzabal, E.; Priego, R. Low-cost platforms used in Control Education: An educational case study. IFAC Proceedings Volumes. 2013, 46(17), 256-261.
[9] Reguera, P.; García, D.; Domínguez, M.; Prada, M. A.; Alonso, S. A low-cost open source hardware in control education. case study: Arduino-feedback ms-150. IFAC-PapersOnLine. 2015, 48(29), 117-122.
[10] Ebrahim, A. Lab kit for liquid level measurement using valve opening with PID controller: Ahmed Abdulhameed Ebrahim, Yvette Shaan-Li Susiapan. TSSA. 2020, 3.
[11] Oteri, O. M. The arduino e-kit as applied in engineering, science and technology e-learning. 2020 Sixth International Conference on e-Learning (econf), Sakheer, Bahrain , 6-7 Dec. 2020; IEEE: Manhattan, New York, USA.
[12] Oliver, J. P.; Haim, F. Lab at home: Hardware kits for a digital design lab. IEEE Trans. Educ. 2008 52(1), 46-51.

Reviewer 3 Report
The paper proposes a low-cost lab. Kit at home called Lab-Tec@Home and evaluates it with 290 students through their feedback.
The Lab-Tec@Home kit looks a simple kit to support students who studies at their home with their own computing devices. The kit composes of ordinary components, but those components are widely used to teach Embedded Systems subject.
- The major limitation(serious flaws) of this paper is that the design and analysis of evaluation for the kit are informally performed.
(1) As for evaluation design, authors describe only the student group with the kit. The paper should describe the other group, i.e., the group without the kit. In addition, authors should give that the conditions of both groups are the same or fair, not biased.
(2) As for analysis, authors describe qualitative criteria for assessment and provide scores of each criterion. For assuring reliability internal consistency of the assessment results, thorough statistical analysis is required, i.e., hypotheses, reasonable data collection, sample characteristics, formal analysis method, and so on.
- Table 3 shows the comparison between the proposed kit and previous kits. In my opinion, the effectiveness is more important than the cost (even though I understand the cost is one of major issues in education). The comparison of effectiveness between the proposed kit and previous kits should be provided for insisting the educational effectiveness of the proposed kit.
Figure 2 does not give any meanings, because the distribution of individual student’s residence area does not have any influences on the evaluation of the Lab-Tec@Home.
So does Figure 6. What does Figure 6 exist for?
Abstract should be modified to include the assessment results.
The description of five primary points (p.4) should be improved with consistent way.
Authors should elaborate references from 42 through 45. They are referred in the section 4.4, but the explanations are too short.
Author Response
Dear Reviewer 3, please find below our detailed responses to your comments and suggestions. Our response is always in italic. Changes in revised manuscript are highlighted in yellow in the attached document.
Reviewer 3:
The paper proposes a low-cost lab. Kit at home called Lab-Tec@Home and evaluates it with 290 students through their feedback.
The Lab-Tec@Home kit looks a simple kit to support students who studies at their home with their own computing devices. The kit composes of ordinary components, but those components are widely used to teach Embedded Systems subject.
- First comment
The major limitation(serious flaws) of this paper is that the design and analysis of evaluation for the kit are informally performed.
Thank you for your observation. We apologize for any misunderstanding. A paragraph explaining the experiment setup is presented in Subsection 2.4 to explain the characteristics selected by each group selected by the study.
a) As for evaluation design, authors describe only the student group with the kit. The paper should describe the other group, i.e., the group without the kit. In addition, authors should give that the conditions of both groups are the same or fair, not biased.
Thank you for your observation. To attend this comment an analysis based on the scores presented on the group with kit and without kit are presented in Section 5. Assessment of student’s performance. Furthermore, a discussion of the effects on the ECOA and student’s scores is presented to justify the findings on this paper.
Taking into account your comment, some sentences in Section 5 (page 13, lines 386-419), Fig. 13 and Fig. 14 were added for the revised manuscript.
b) As for analysis, authors describe qualitative criteria for assessment and provide scores of each criterion. For assuring reliability internal consistency of the assessment results, thorough statistical analysis is required, i.e., hypotheses, reasonable data collection, sample characteristics, formal analysis method, and so on.
Thank you for your remark. The data recollection is presented based on the experiment setup from Subsection 2.4, it is presented a distribution of student’s scores in both cases the group with the kit (experimental groups) and the groups without kit (control groups). The scores are presented and explained how the kit impacts on the results on the students and their learning experiences over the course. In many cases it is related highly related with the results the student present on the perception of the course on the ECOA.
- Second comment
Table 3 shows the comparison between the proposed kit and previous kits. In my opinion, the effectiveness is more important than the cost (even though I understand the cost is one of major issues in education). The comparison of effectiveness between the proposed kit and previous kits should be provided for insisting the educational effectiveness of the proposed kit.
Thank you for your remark. We apologize for any misunderstanding. Lab kits often require extensive development time to design the hardware, software, and lab exercises [12]. Nevertheless, as it is well mention in [1], when lab kits are used for control engineering education, the major drawback is that they are quite expensive, and thus not affordable by many educational institutions. Table 3 shows the feasibility to carried out the proposed lab kit; then, using less number of components and an electronic platform with a freely available software, students can test and validate control engineering theoretical concepts.
In order to evaluate the educational effectiveness of the proposed lab kit for control engineering courses, the final scores of students, using and without using the lab kit, are presented for the revised manuscript in Section 5 (page 13, lines 386-419) and references [1], [12] were added.
a) Figure 2 does not give any meanings, because the distribution of individual student’s residence area does not have any influences on the evaluation of the Lab-Tec@Home.
Authors appreciate the Reviewer’s comment. Thank you. The aim of the proposed lab kit is to expand the access of hands-on control education at the undergraduate level during a critical COVID-19 pandemic situation. Hence, Figure 2 shows the influence of the lab kit in Mexico. As it can be seen, students from different states, whose attended a totally remote course, were able to obtain the lab kit due to its low cost and less number of components required.
We apologize for any misunderstanding. Some sentences were added for the revised manuscript (Section 2, page 3, lines 126-127).
b) So does Figure 6. What does Figure 6 exist for?
The Reviewer raises an interesting concern, thank you for your observation. In this work, two main software are adopted: Arduino programs are used to handle the problem of the real-time data recording and carry out feedback control experiments. Moreover, students performed graphics using MATLAB/SimulinkTM to analyze data, conduct a model parameter estimation, and design the controller. To test the lab kit in open loop an input step is applied. Thus, using the real data collected by the Arduino, a plot is carried out in MATLABTM, in which student perform model identification by estimating the First Order Plus Time Delay (FOPTD) parameters [A]. Figure 6 presents the real system output and the simulation output using the three different FOPTD graphical methods (Ziegler-Nichols, Miller and Analytic method). Therefore, the student can verify that there exists a good agreement among two responses (Miller and Analytic method), indicating that the model performs well.
Furthermore, we apologize for any misunderstanding, and some sentences have been modified in (page 8, lines 247-251) to the revised manuscript in order to avoid this.
c) Abstract should be modified to include the assessment results.
Authors do appreciate the Reviewer’s observation. Based in this, the Abstract (page 1, in the previous manuscript) has been modified for the revised manuscript.
d) The description of five primary points (p.4) should be improved with consistent way.
Thank you for your observation. Based on the Reviewer’s suggestion some sentences in Subsection 2.2, page 5, lines 154-164) were added for the revised manuscript.
e) Authors should elaborate references from 42 through 45. They are referred in the section 4.4, but the explanations are too short.
Authors appreciate Reviewer’s comment, thank you. We apologize for any misunderstanding. Based on the Reviewer’s suggestion, for the revised manuscript authors added some sentences in (page 13, lines 379-385, and page 15, lines 434-437).
We believe that these corrections and additions adequately address all the points that have been raised by the Reviewers. Authors also believe that the comments significantly helped to improve the manuscript.
We hope that the paper can be accepted for publication.
On behalf of all authors,
Carlos Sotelo, Ph.D.
Professor
Mechatronics and Electrical Department,
Sciences and Engineering School,
Tecnologico de Monterrey, Monterrey Campus
64849 Monterrey Nuevo León, México.
Email: carlos.sotelo@tec.mx
References:
[1] de Moura-Oliveira, P. B; Hedengren, J. D.; Rossiter, J. A. Introducing Digital Controllers to Undergraduate Students using the TCLab Arduino Kit. IFAC-PapersOnLine. 2020, 53(2), 17524-17529.
[2] Rossiter, J. A.; Pope, S. A.; Jones, B. L.; Hedengren, J. D. Evaluation and demonstration of take home laboratory kit. IFAC-PapersOnLine. 2019, 52(9), 56-61. https://doi.org/10.1016/j.ifacol.2019.08.124.
[3] Sarik, J.; Kymissis, I. Lab kits using the Arduino prototyping platform. Proceedings - Frontiers in Education Conference (FIE), Arlington, VA, USA, 27-30 Oct. 2010; IEEE: Manhattan, New York, USA.
[4] Alvarado, I.; Maestre, J. M. A Lightsaber to Introduce Students to Microcontrollers. IFAC-PapersOnLine. 2019. 52(9), 139-143.
[5] Oltean, S. E. Mobile robot platform with Arduino uno and raspberry pi for autonomous navigation. Procedia Manuf. 2019. 32, 572-577.
[6] KüçükaÄŸa, Y.; Facchin, A.; Torri, C.; Kara, S. An original Arduino-controlled anaerobic bioreactor packed with biochar as a porous filter media. MethodsX. 2022, 101615.
[7] Prima, E. C.; Karim, S.; Utari, S.; Ramdani, R.; Putri, E. R. R.; Darmawati, S. M. Heat transfer lab kit using temperature sensor based arduinoTM for educational purpose. Procedia Eng. 2017, 170, 536-540.
[8] Irigoyen, E.; Larzabal, E.; Priego, R. Low-cost platforms used in Control Education: An educational case study. IFAC Proceedings Volumes. 2013, 46(17), 256-261.
[9] Reguera, P.; García, D.; Domínguez, M.; Prada, M. A.; Alonso, S. A low-cost open source hardware in control education. case study: Arduino-feedback ms-150. IFAC-PapersOnLine. 2015, 48(29), 117-122.
[10] Ebrahim, A. Lab kit for liquid level measurement using valve opening with PID controller: Ahmed Abdulhameed Ebrahim, Yvette Shaan-Li Susiapan. TSSA. 2020, 3.
[11] Oteri, O. M. The arduino e-kit as applied in engineering, science and technology e-learning. 2020 Sixth International Conference on e-Learning (econf), Sakheer, Bahrain , 6-7 Dec. 2020; IEEE: Manhattan, New York, USA.
[12] Oliver, J. P.; Haim, F. Lab at home: Hardware kits for a digital design lab. IEEE Trans. Educ. 2008 52(1), 46-51.

Reviewer 4 Report
The submitted manuscript deals with a Lab-Tec@Home: A cost-effective kit for online control engineering education, which might be related to the main focus of this journal. However, it has still some flaws I can find out as the below:
Please observe basic requirements of journal „Guide for Authors" (GfA) on the formatting of manuscript components. The current version is not satisfying with the basic guidelines that are not acceptable, especially the section of references.
The introduction and related works are too short thar are necessary to add more information. Current stance can’t support the authors’ claims due to not having enough background supports of the previous researchers.
There are figures and tables for the study, which are redundant information, and the deep data analysis is required to improve the quality and readability of manuscript.
This research is based on response data. Hence it is important to discuss in detail.
The discussion on results is poorly presented. The execution of the proposed methodology is appreciable while the discussion of the obtained results must be well improved highlighting the insights of the research findings and with support from earlier literature. I find many unsupported statements in this section.
Please revise your conclusion part into more details. Basically, you should enhance your contributions, limitations, underscore the scientific value added of your paper, and/or the applicability of your findings/results and future study in this session. Also, please avoid putting numberings in the various sections.
References are not formatted to satisfy GfA requirements.
There are too many incorrect language details in your manuscript (lexical, grammatical and spelling errors, and phrases that do not belong to correct English). Particularly, the authors have used many long sentences that are really difficult to follow and understand.
Author Response
Dear Reviewer 4, please find below our detailed responses to your comments and suggestions. Our response is always in italic. Changes in revised manuscript are highlighted in yellow in the attached document.
Reviewer 4:
The submitted manuscript deals with a Lab-Tec@Home: A cost-effective kit for online control engineering education, which might be related to the main focus of this journal. However, it has still some flaws I can find out as the below.
- First comment
Please observe basic requirements of journal „Guide for Authors" (GfA) on the formatting of manuscript components. The current version is not satisfying with the basic guidelines that are not acceptable, especially the section of references.
Thank you for your remark. We apologize for the mistake. Based on this, authors have carefully checked the Guide for Authors and format of references was improved for the revised manuscript.
- Second comment
The introduction and related works are too short thar are necessary to add more information. Current stance can’t support the authors’ claims due to not having enough background supports of the previous researchers.
Authors appreciate Reviewer’s comment, thank you. Based on the Reviewer’s suggestion, for the revised manuscript authors added the references [1][8][9][10] in which lab kit options are used for teaching control engineering education. However, due to the originality of the present work where a cost-effective lab kit is designed and implemented to expand the access of hands-on control education at the undergraduate level during a critical COVID-19 pandemic situation, no comparable results were found in literature field.
Furthermore, taking into account your comment, some sentences in (page 2, lines 62-64 and lines 66-68) were modified for the revised manuscript (page 2, lines 50-54), and references [1][8][9][10] were added for the revised manuscript.
3. Third comment
There are figures and tables for the study, which are redundant information, and the deep data analysis is required to improve the quality and readability of manuscript.
Authors do appreciate the Reviewer’s suggestion. Based on this, Figure 7 and Table 2 (page 7, in the previous manuscript) have been removed for the revised manuscript.
4. Fourth comment
This research is based on response data. Hence it is important to discuss in detail.
Thank you for your observation. We apologize for any misunderstanding. Based on your comment, a discussion and analysis of the student’s scores of different courses and the analysis of the ECOA are presented in Section 5 to support the results of the experiment.
Taking into account your comment, some sentences in Section 5 (pages 13-14, lines 386-419), Fig. 13 and Fig. 14 were added for the revised manuscript.
- Fifth comment
The discussion on results is poorly presented. The execution of the proposed methodology is appreciable while the discussion of the obtained results must be well improved highlighting the insights of the research findings and with support from earlier literature. I find many unsupported statements in this section.
Thank you for your remark. To attend this comment an analysis based on the scores presented on the group with kit and without kit are added. Moreover, a discussion of the effects on the ECOA and student’s scores is presented to justify the findings on this paper. Thus, using the lab kit the learning experience is improved, and it is highly related with the results that the student present on the perception of the course on the ECOA.
Taking into account your comment, some sentences in Section 5 (pages 13-14, lines 386-419), Fig. 13 and Fig. 14 were added for the revised manuscript.
6. Sixth comment
Please revise your conclusion part into more details. Basically, you should enhance your contributions, limitations, underscore the scientific value added of your paper, and/or the applicability of your findings/results and future study in this session. Also, please avoid putting numberings in the various sections.
The Reviewer raises an interesting concern, thank you for your observation. Prior the designing of the proposed educational project a research has been done on different types of lab kit used for engineering education [1] [3][4][5][6][7][8][9][10]. However, authors found that some of the research works have limitations. Thus, in order to highlight the strength points of the proposed research work, authors present the following main contributions:
- Easy-to-use hardware/software solution for designing take-home lab exercises. To encourage the widespread adoption of distributed lab kits, the proposed lab kit is small size, portable, and ready to be used. Moreover, the software to use the kit is freely available both for Python and MATLAB/Simulink, and supporting educational videos and material are freely made available. Thus, the kit replaces expensive equipment with an affordable alternative that can be easily shipped anywhere in the world and used by students who do not have a formal laboratory space to use [11][12].
- Useful for teaching PID controller design at the undergraduate level. Considering PID type controllers have been commonly used in industrial applications, this lecture topic has a high relevance in most introductory control courses. However, students facing lack of knowledge in controlling the PID as a hardware where this is due to the complexity of the PID controller to be implemented [10]. Thus, using open hardware/software and based on a real situation, the proposed system provides an effective and attractive solution for teaching PID controller design at the undergraduate level.
- Appropriate to be implemented in remote course. Unlike previous research works, where lab kits are used in classroom, in the present work the kit is implemented for massive remote courses. In addition, based on an institutional student survey (ECOA), the results were quite positive where most of the students indicated that although the class was taught totally remote, they pick up practical skills which makes them feel motivated and inspired to give the best effort to carried out the tasks.
Thus, the proposed lab kit delivers a consistent, high quality laboratory experience for students whose participate in distance education.
On the other hand, as previous research works [1][8][9][10] used for control engineering education, the main limitation of the proposed research work is that the lab kit is useful for linear time invariant and single input – single output systems. For nonlinear and multivariable processes mentioned in [42][45] (in the previous manuscript), new lab kits will be carefully designed to teach advanced control strategies such as nonlinear and model predictive control.
Taking into account your comment, and to highlight the strength points and mention the limitation of this research work, some sentences in (page 13, lines 379-385, and page 15, lines 434-437), and references [1][3][4][5][6][7] [8][9][10][11][12] were added for the revised manuscript.
- Seventh comment
References are not formatted to satisfy GfA requirements.
Thank you for your remark. Format of references was improved for the revised manuscript.
8. Eight comment
There are too many incorrect language details in your manuscript (lexical, grammatical and spelling errors, and phrases that do not belong to correct English). Particularly, the authors have used many long sentences that are really difficult to follow and understand.
Thank you for your observation. English writing was thoroughly checked and modified for the revised manuscript.
We believe that these corrections and additions adequately address all the points that have been raised by the Reviewers. Authors also believe that the comments significantly helped to improve the manuscript.
We hope that the paper can be accepted for publication.
On behalf of all authors,
Carlos Sotelo, Ph.D.
Professor
Mechatronics and Electrical Department,
Sciences and Engineering School,
Tecnologico de Monterrey, Monterrey Campus
64849 Monterrey Nuevo León, México.
Email: carlos.sotelo@tec.mx
References:
[1] de Moura-Oliveira, P. B; Hedengren, J. D.; Rossiter, J. A. Introducing Digital Controllers to Undergraduate Students using the TCLab Arduino Kit. IFAC-PapersOnLine. 2020, 53(2), 17524-17529.
[2] Rossiter, J. A.; Pope, S. A.; Jones, B. L.; Hedengren, J. D. Evaluation and demonstration of take home laboratory kit. IFAC-PapersOnLine. 2019, 52(9), 56-61. https://doi.org/10.1016/j.ifacol.2019.08.124.
[3] Sarik, J.; Kymissis, I. Lab kits using the Arduino prototyping platform. Proceedings - Frontiers in Education Conference (FIE), Arlington, VA, USA, 27-30 Oct. 2010; IEEE: Manhattan, New York, USA.
[4] Alvarado, I.; Maestre, J. M. A Lightsaber to Introduce Students to Microcontrollers. IFAC-PapersOnLine. 2019. 52(9), 139-143.
[5] Oltean, S. E. Mobile robot platform with Arduino uno and raspberry pi for autonomous navigation. Procedia Manuf. 2019. 32, 572-577.
[6] KüçükaÄŸa, Y.; Facchin, A.; Torri, C.; Kara, S. An original Arduino-controlled anaerobic bioreactor packed with biochar as a porous filter media. MethodsX. 2022, 101615.
[7] Prima, E. C.; Karim, S.; Utari, S.; Ramdani, R.; Putri, E. R. R.; Darmawati, S. M. Heat transfer lab kit using temperature sensor based arduinoTM for educational purpose. Procedia Eng. 2017, 170, 536-540.
[8] Irigoyen, E.; Larzabal, E.; Priego, R. Low-cost platforms used in Control Education: An educational case study. IFAC Proceedings Volumes. 2013, 46(17), 256-261.
[9] Reguera, P.; García, D.; Domínguez, M.; Prada, M. A.; Alonso, S. A low-cost open source hardware in control education. case study: Arduino-feedback ms-150. IFAC-PapersOnLine. 2015, 48(29), 117-122.
[10] Ebrahim, A. Lab kit for liquid level measurement using valve opening with PID controller: Ahmed Abdulhameed Ebrahim, Yvette Shaan-Li Susiapan. TSSA. 2020, 3.
[11] Oteri, O. M. The arduino e-kit as applied in engineering, science and technology e-learning. 2020 Sixth International Conference on e-Learning (econf), Sakheer, Bahrain , 6-7 Dec. 2020; IEEE: Manhattan, New York, USA.
[12] Oliver, J. P.; Haim, F. Lab at home: Hardware kits for a digital design lab. IEEE Trans. Educ. 2008 52(1), 46-51.

Round 2
Reviewer 2 Report
The authors improved the manuscript according with the recommendations.
Reviewer 3 Report
I have no further comments.
Reviewer 4 Report
First of all, I appreciate to the authors for making efforts to carry out the changes by the referees. I think the authors did a good job in clarifying the queries that this manuscript is substantially improved.